# Public stigma associated with mental illnesses in Pakistani university students: a cross sectional survey

Ahmed Waqas[1], Muhammad Zubair[2], Hamzah Ghulam[2], Muhammad Wajih Ullah[2] and Muhammad Zubair Tariq[2]

[1] CMH Lahore Medical College and Institute of Dentistry, Lahore Cantt, Pakistan
[2] FMH College of Medicine and Dentistry, Lahore, Pakistan

## ABSTRACT

**Background.** The objectives of the study were to explore the knowledge and attitudes of Pakistani university students toward mental illnesses. People with mental illnesses are challenged not only by their symptoms but also by the prejudices associated with their illness. Acknowledging the stigma of mental illness should be the first essential step toward devising an appropriate treatment plan.

**Methods.** A cross-sectional survey was conducted at the University of Punjab, Lahore, CMH Lahore Medical and Dental College, Lahore, and University of Sargodha, Sub-campus Lahore, from February to May 2014. The self-administered questionnaire consisted of three sections: demographics, general knowledge of psychiatric illnesses, and Community Attitudes towards Mental Illnesses (CAMI) Scale. The questionnaire was distributed to 650 participants enrolled in different disciplines (Social Sciences, Medicine and Formal Sciences).

**Results.** Response rate was 81% (527/650 respondents). Mean age was 20.98 years. Most of the students (331, 62.8%) had an urban background and studied Social Sciences (238, 45.2%). Four hundred and eighteen respondents (79.3%) considered religion very important and most respondents considered psychiatrists (334, 63.4%) and spiritual leaders (72, 13.7%) to be best able to treat mental illnesses. One hundred and sixty nine respondents (32.1%) considered black magic to be a cause of mental illness. Only 215 (41%) respondents had ever read an article on mental illnesses. Multiple regression analysis revealed study discipline, exposure, perceived causes of mental illnesses and superstitions to be significantly associated with attitudes towards mental illnesses ($p < .05$).

**Conclusion.** Although low awareness and exposure were found in this sample of Pakistani university students, their attitude towards mental illnesses was generally positive. Most respondents gave supernatural explanations for mental illnesses but only a few believed that spiritual leaders can play a role in treatment.

Corresponding author
Ahmed Waqas,
ahmedwaqas1990@hotmail.com

## INTRODUCTION

In recent years much has been published about the stigma attached to mental illnesses. However, most work to date has focused on western populations, and there is a dearth of

research literature on stigma in the developing world. People with mental illness are one of the most stigmatized strata of our society. According to Elliot and colleagues, public stigma associated with mental illnesses renders the mentally ill socially illegitimate. They are perceived as incapable of normal interaction, dangerous and unpredictable, and these perceptions lead to their exclusion from the community (*Elliott et al., 1982*). Therefore, they are challenged not only by their illness but also by the stigma and stereotypes associated with them by the community.

In Pakistan, it is estimated that neuropsychiatric disorders account for 11.9% of the overall burden of disease (*WHO, 2008*). In developing countries, less than 35% of psychiatric patients receive care (*WHO, 2001a*). Despite the availability of psychiatric care (although meager), individuals often do not seek basic treatment due to their fear of public stigma and shame. Therefore, psychiatrists who treat people with mental illnesses have been encouraged to address the social context, nature of adverse experiences, self-image and attitudes of the community. Acknowledging negative attitudes towards mental illness should be the first step in devising appropriate mental health policies and treatment plans (*Byrne, 2000*). Failure to recognize public stigma and its effects can lead to isolation of people with mental illness from the community (*Goffman, 2009*). Furthermore, people with mental illness who justify and accept prejudice experience self-stigma, which can lead to stress and can compromise their capability for independent living (*Corrigan, Patrick & Penn, 1999*). Their increased stress may also cause psychological problems such as depression (*Link et al., 1997*), anxiety (*Farina, 1981*) and low self-esteem (*Link, 1987*).

Pakistan is a culturally and ethnically diverse country that is home to a number of religious branches of Islam whose practitioners can nurture very complex belief systems. Beliefs in black magic, the evil eye and possession by Jinni (demons) are prevalent in this society (A Gadit & T Callanan, pers. com., 2006). Spiritual leaders are revered and attract huge numbers of followers and devotees to their shrines. The tendency to turn to spiritual resources has emerged as an effective coping mechanism for various issues in this part of the world (*Voll, 1992*). Apart from spiritual leaders, a huge number of shamans with no formal qualifications have also emerged. This belief system of Pakistani society has implications for the knowledge of, and stereotypes attached to, mental illnesses. As a result, people in this part of the world, stricken by low literacy rates, poor socioeconomic conditions and prevalent stigma, are hugely dependent on shamans for the treatment of mental illnesses. Among the widely accepted causes of mental illnesses are possession by demons and magical spells cast by enemies, and among the accepted treatments are talismans, amulets and incantations (*Gadit & Reed, 2004*).

The paucity of knowledge about public stigma, stereotypes and superstitions prevailing in Pakistani society warranted this study, which was designed with three aims: (1) to assess the prevalence of public stigma associated with mental illnesses in a sample of university students in Pakistan, (2) to determine the reasons for these stigmata, and (3) to determine the prevalence of supernatural beliefs and their effect on stigmata associated with mental illnesses.

## METHODS

This cross-sectional survey was conducted in February 2014 at the University of Punjab, Lahore, CMH Lahore Medical and Dental College, Lahore and the University of Sargodha, Sub-campus Lahore. It was approved by the Ethics Review Committee of CMH Lahore Medical College and Institute of Dentistry. An anonymous, self-administered questionnaire was distributed (convenience sampling) to 650 students enrolled in degree programs in various disciplines (Social Sciences, Medicine and Formal Sciences) who were willing to participate in the survey. Written informed consent was provided by each participant. They were informed about the objectives of the survey and ensured anonymity and that only group-level (not individual) findings would be reported.

The survey questionnaire consisted of three sections: demographic information, a section assessing their knowledge of mental illnesses, and the Community Attitudes toward the Mentally Ill (CAMI) Scale developed by *Taylor & Dear (1981)*. The first section recorded the participants' demographic information. The second section assessed whether they had any prior knowledge of and exposure to mental illnesses. Three questions were asked: (1) Who can best cure mental illnesses? (A) General physician, (B) Psychiatrist, (C) Spiritual leader (D) Other. (2) Have you ever talked with a person with a mental illness, read books or articles about mental illness, or cared for or had any relatives with mental illness? (3) What do you think are the causes of mental illness? The participants indicated possible causes from a table listing various causes of mental illnesses and superstitions. Multiple responses were allowed.

The CAMI scale measures stigma and attitudes of the community towards mental illnesses. It defines mental illness as referring to people needing treatment for mental disorders but who are capable of independent living outside a hospital (*Taylor & Dear, 1981*). This scale consists of 40 statements with a Likert scale type of response. The four subscales assess four types of attitudes towards mental illness. "Authoritarianism" reflects oppressive attitudes towards the mentally ill, "Benevolence" reflects a sympathetic attitude, "Social restrictiveness" considers the mentally ill as a threat to the society, and "Community mental health ideology (CMHI)" supports the idea of community-oriented care for the mentally ill (*Taylor & Dear, 1981*). Each subscale comprises 10 items with 5 positively scored and 5 negatively scored items. Scores are reversed on negatively scored items and then the total score for each subscale is calculated. The maximum score for each subscale is 50.

The data was analyzed with SPSS v. 20 software. The chi-squared test and Cramer's V was used to find associations between demographic variables, prevalent superstitions, study discipline, exposure and whether or not respondents believed that shamans could play a therapeutic role in treating mental illness. Multiple regression analysis (backward method) was used to identify associations between demographic variables, exposure, knowledge of the true causes of mental illness, prevalent superstitions and scores on each of the CAMI subscales.

For regression analysis, categorical variables with more than two categories, such as background, importance of religion and study discipline, were coded as dummy variables

with 0 and 1 as coding values. A histogram was plotted to visualize the distribution of the data as normal or non-normal, Probability–probability (P–P) plots, the Durbin–Watson diagnostic test and colinearity diagnostics were run to ensure that the assumptions of regression analysis were not violated. The level of significance was set at $p < .05$.

## RESULTS

### Demographics

The response rate was 81.1% (527 respondents out of 650 students who received the questionnaire). Mean age of the respondents was 20.98 years (2.66), 312 (59.2%) were females and 215 (40.8%) males. Most students were enrolled in a Social Sciences degree program (238, 45.2%), followed by Medicine (202, 38.3%) and Formal Sciences (87, 16.5%). 331 (62.8%) of the students came from an urban background, 124 (23.5%) from a rural background and 72 (13.7%) from a semi-urban background. 418 (79.3%) considered religion very important, 102 (19.4%) considered it important and only 7 (1.3%) felt religion was unimportant.

### Exposure to mental illnesses

334 (63.38%) participants believed that psychiatrists are best able to cure mental illness, followed by general physicians (79, 14.99%), shamans or spiritual leaders (72, 13.66%) and others (42, 7.97%). 215 (40.8%) participants had ever read an article or book on mental illness. Only 273 (51.8%) had ever talked to a person with mental illness, and 237 (45.0%) had ever cared for a person with mental illness. 276 (52.4%) had a relative with a mental illness. Chi-squared tests revealed a significant association between having cared for the mentally ill (chi-squared $= 4.56$) and the study discipline (chi-squared $= 6.27$) and punishment from God (chi-squared $= 3.5$) as a cause of mental illness, and whether or not shamans or spiritual leaders were believed to be best able to cure mental illnesses (all $p < .05$). According to these results, respondents who had cared for someone with a mental illness, were enrolled in the non-medical study discipline, and considered punishment from God as a cause of mental illness had a higher tendency to report that spiritual leaders were best able to cure mental illnesses. No association was found between other superstitious beliefs and whether spiritual leaders were best able to cure mental illnesses ($p > .05$).

### Prevalent superstitions and knowledge of true psychopathologies

118 (22.4%) students believed in the evil eye, 169 in black magic (32.1%), 135 (25.6%) in punishment from God and 134 (25.4%) in demonic possession as a cause of mental illness. The frequency distribution of true psychopathologies as reported by participants was as follows: drug abuse, 278 (52.8%); psychosocial trauma, 267 (50.7%); alcohol abuse, 180 (34.2%); work-related stress, 301 (57.1%); genetic predisposition, 211 (40%); physical abuse, 241 (45.7%); poverty, 184 (34.9%); study-related stress, 236 (44.8%); and divorce, 260 (49.3%). Chi-squared tests revealed significant associations between study disciplines (Medicine vs. Non-medicine), superstitions and true psychopathologies. Only those variables that yielded significant associations are shown in Table 1. This table reflects the finding that medical students were more likely to report psychosocial trauma, work-related

**Table 1 Association between study discipline and beliefs in causes of mental illness.**

| Causes of mental illness | Chi-squared value | Degrees of freedom (df) | Cramer's V |
|---|---|---|---|
| Trauma | 165.19[**] | 1 | .560[**] |
| Work stress | 21.57[**] | 1 | .202[**] |
| Genetic predisposition | 34.54[**] | 1 | .256[**] |
| Physical abuse | 59.20[**] | 1 | .335[**] |
| Study-related stress | 20.28[**] | 1 | .196[**] |
| Divorce | 27.92[**] | 1 | .198[**] |
| Evil eye | 6.96[*] | 1 | .230[*] |
| Punishment from God | 18.14[**] | 1 | .186[**] |
| Demonic possession | 10.77[**] | 1 | .143[**] |

**Notes.**
[*] $p$ value $<.05$.
[**] $p$ value $<.001$.

**Table 2 Association between background and belief in superstitious causes of mental illnesses.**

| Variable | Chi-squared value | Degrees of freedom (df) | Cramer's V |
|---|---|---|---|
| Punishment from God | 22.66[**] | 1 | .21[**] |
| Demonic possession | 10.1[*] | 1 | .14[*] |

**Notes.**
[*] $p < .01$.
[**] $p < .001$.

stress, genetic predisposition towards mental illnesses, physical abuse, study-related stress and divorce as causes of mental illness. In contrast, non-medical students were more likely to report the evil eye, punishment from God and possession by Jinni (demons) as possible causes of mental illnesses.

## Background and belief in superstitions

The background of participants also showed a statistically significant association with superstitions (Table 2). Participants with a rural background were more likely to identify superstitions as the cause of mental illness.

## CAMI subscale scores and determinants

The median scores of participants on CAMI subscales were Authoritarian 29, Benevolence 36, Social restrictiveness 28 and Community mental health attitude (CMHI) 32.5. Multiple regression analysis (backward method) was run for the subscale scores with demographic variables, exposure, superstitions and true psychopathologies as variables predictive of the variance in scores on the different subscales. Only the final models are reported in Tables 3–6.

## DISCUSSION

The study sample consisted of undergraduate students enrolled in three universities in Lahore and therefore reflects the attitudes of the literate strata of Pakistani society. Overall,

**Table 3 Predictors of Authoritarian subscale.** Multiple regression analysis for the CAMI Authoritarian subscale.

| Predictor | B | Standard error of B | β |
|---|---|---|---|
| Constant | 28.876 | .487 | |
| Study discipline | −1.894 | .361 | −.226[**] |
| Ever read | −1.204 | .341 | −.145[**] |
| Drug abuse | .677 | .334 | .083[*] |
| Genetic predisposition | −.698 | .352 | −.084[*] |
| Punishment from God | 1.333 | .383 | .143[**] |
| Religion | 1.069 | .414 | .106[*] |

Notes.

Adjusted $R^2 = .159$.

[*] $p < .05$.

[**] $p < .001$.

**Table 4 Predictors of Benevolence subscale.** Multiple regression analysis for the CAMI Benevolence subscale.

| Predictor | B | Standard error of B | β |
|---|---|---|---|
| Constant | 34.142 | .467 | |
| Study discipline | 2.363 | .534 | .209[***] |
| Drug abuse | 1.039 | .452 | .095[*] |
| Trauma | 2.248 | .525 | .205[***] |
| Alcohol | −1.382 | .490 | −.119[**] |
| Work-related stress | 1.077 | .435 | .097[*] |
| Physical abuse | .864 | .483 | .078[*] |
| Evil eye | −1.896 | .509 | −.144[***] |
| Poverty | −1.429 | .461 | −.124[**] |
| Background (rural vs. other) | −1.256 | .515 | −.097[*] |

Notes.

Adjusted $R^2 = .251$.

[*] $p < .05$.

[**] $p < .01$.

[***] $p < .001$.

the participants had favorable attitudes towards people with mental illness, as reflected by their scores on the CAMI subscales. They scored lower on the Authoritarian and Social restrictiveness subscales and higher on the Benevolence and CMHI subscales. Comparatively, medical students had more positive attitudes towards people with mental illness than students enrolled in nonmedical study disciplines. Students who had read books or articles, cared for or talked with people with mental illness were less authoritative and less socially restrictive, more benevolent, and had a more favorable community attitude. Behavioral sciences are an integral part of the undergraduate medical curriculum in Pakistani medical schools. Accordingly, most medical students in this survey already had a better knowledge of mental illnesses than students in other disciplines. Moreover, most medical students were already involved in clinical training, so their exposure to people with mental illnesses

**Table 5 Predictors of social restrictiveness subscale.** Multiple regression analysis for the CAMI Social restrictiveness subscale.

| Predictor | B | Standard error of B | β |
|---|---|---|---|
| Constant | 29.067 | .324 | |
| Ever talked | −.757 | .335 | −.097[*] |
| Ever cared | −.550 | .333 | −.070[*] |
| Trauma | −1.651 | .357 | −.211[**] |
| Genetic | −.622 | .362 | −.078[*] |
| Evil eye | .743 | .408 | .079[*] |
| Poverty | .866 | .350 | .105[*] |
| Punishment from God | .871 | .388 | .097[*] |

Notes.

Adjusted $R^2 = .107$.

[*] $p < .05$.

[**] $p < .001$.

**Table 6 Predictors of CMHI subscale.** Multiple regression analysis for the CAMI CMHI subscale.

| Predictor | B | Standard error of B | β |
|---|---|---|---|
| Constant | 30.733 | .536 | |
| Study discipline | 1.657 | .446 | .185[**] |
| Ever read | .795 | .362 | .090[*] |
| Trauma | 1.821 | .425 | .209[**] |
| Poverty | −.931 | .374 | −.102[*] |
| Punishment by God | −1.044 | .418 | −.105[*] |
| Demonic possession | −.909 | .414 | −.091[*] |

Notes.

Adjusted $R^2 = .175$.

[*] $p < .05$.

[**] $p < .0014$.

was greater. This may have led medical students to adopt more lenient attitudes towards people with mental illnesses compared to students in nonmedical disciplines.

Overall, participants had a poor knowledge of the biopsychosocial causes of mental illness. This finding is consistent with other studies carried out in England (*Evans-Lacko, Henderson & Thornicroft, 2013*) from 2009 to 2012 and in Egypt (*Dessoki & Hifnawy, 2009*). Respondents in our study expressed better knowledge of the true causes of mental illnesses and hence more positive attitudes than those reported by *Gureje et al. (2005)*, probably because the present study included university students only rather than the general public. Knowledge of the general public in Pakistan regarding mental illnesses might be similar to that found by Gureje and colleagues.

Many studies have reported a link between perceived causes of mental illnesses and stigmatizing attitudes. In a similar study conducted by *Gureje et al. (2006)*, these causes fell into two domains: biopsychosocial and religio-magical. The former group had more tolerant attitudes towards people with mental illnesses. The present analysis, however,

yielded slightly different results. It may be informative to further subclassify these domains in order to explore in more detail the relationship between knowledge of the causes and attitudes towards mental illness. In contrast to *Gureje et al. (2006)*, participants in the present study who reported substance abuse, alcoholism and poverty as causes of mental illness were generally more authoritative and socially restrictive, less benevolent, and had less favorable community attitudes. Compared to other biopsychosocial causes, substance abuse and alcoholism elicited negative attitudes because of beliefs that alcoholics and drug abusers are dangerous, unpredictable, capable of helping themselves and more responsible for their condition (*Crisp et al., 2000*). These attitudes consequently lead to greater public stigma, negative attitudes and social rejection (*Schomerus et al., 2011*).

Greater religiosity correlated directly with more authoritative attitudes, reflecting oppressive attitudes towards people with mental illnesses. A similar study in Benin reported more authoritative attitudes among members of the Muslim clergy (*Igbinomwanhia, James & Omoaregba, 2013*), in consonance with the potential influence of religious beliefs on the perceived stigma of mental illness. The belief that mental illness might indicate spiritual failure potentiates stigma and may discourage individuals from seeking psychiatric care (*Trice, Pamela & Bjorck, 2006*).

Superstitious beliefs such as black magic, the evil eye, punishment from God and demonic possession as causes of mental illnesses were highly prevalent among the participants in the present study. Generally, these beliefs tended to be associated with less favorable attitudes among university students towards persons with mental illness. Medical students were less likely to report supernatural causes of mental illnesses, and participants from a rural background tended to identify supernatural causes of mental illnesses more frequently than those with an urban or semi-urban background. The former also had less benevolent attitudes towards people with mental illness. In a recent study, rural residents were much more likely to believe in superstitions such as black magic and the healing powers of talismans and Sufi shrines (*Farooq, 2012*). Three reasons were reported for believing in superstitions: experience, tradition and religion. A much smaller percentage of students in the present sample believed that shamans could play a therapeutic role in mental illness. This reflects a general lack of trust among more literate persons in the therapeutic abilities of shamans.

The stigma associated with mental illness is highly prevalent in Pakistani society. To reduce the stigma, barriers to care and the shame associated with seeking psychiatric help, WHO recommends launching mass awareness programs in all countries to raise the public's knowledge and awareness of the frequency, treatment and recovery process for mental disorders (*WHO, 2001b*). These psycho-educational campaigns should be designed according to the needs of specific groups, addressing their attitudes, fear and concerns pertaining to mental illnesses. These educational interventions and appropriate exposure can decrease the stigma associated with mental illnesses not only among undergraduate medical students (*Papish et al., 2013*) but also among students of other disciplines and grades (*Pinfold et al., 2003*).

### Limitations & recommendations

The results of this study cannot be generalized as the population sample was not obtained randomly. The use of a self-administered questionnaire may have led to information bias. The cross-sectional design of this study limits inferences about causality between exposure, perceived causes and attitudes towards mental illness. Religiosity was measured with a single question: How important is religion in your life? For future studies the use of a reliable scale is advisable. Although an item regarding socioeconomic status was included in the questionnaire, 75 respondents left it blank. Therefore, this variable was excluded from the analysis.

## CONCLUSION

Although overall awareness of and exposure to mental illness were low in this sample of university students, their attitudes towards mental illnesses were generally positive. Most students expressed a belief in supernatural explanations for mental illnesses, whereas only a few believed that spiritual leaders can play a role in their treatment.

## ACKNOWLEDGEMENTS

The authors thank Aroosa Allah Yar and Arooj Allah Yar, students at Hailey College of Commerce, Punjab University, Lahore, Pakistan, and Abeer Asfaq, Azfar Hameed, Hamza Bukhari, Zahra Malik and Nur Ghani, students at CMH Lahore Medical and Dental College, for helping with data collection for this project. They would also like to thank K Shashok (AuthorAID in the Eastern Mediterranean) for improving the use of English in the manuscript.

### Funding

The authors declare there was no funding for this work.

### Competing Interests

The authors declare there are no competing interests.

### Author Contributions

- Ahmed Waqas and Muhammad Zubair conceived and designed the experiments, performed the experiments, analyzed the data, contributed reagents/materials/analysis tools, wrote the paper, prepared figures and/or tables, reviewed drafts of the paper.
- Hamzah Ghulam performed the experiments, contributed reagents/materials/analysis tools, wrote the paper, prepared figures and/or tables, reviewed drafts of the paper.
- Muhammad Wajih Ullah and Muhammad Zubair Tariq performed the experiments, contributed reagents/materials/analysis tools, wrote the paper, reviewed drafts of the paper.

## Human Ethics

The following information was supplied relating to ethical approvals (i.e., approving body and any reference numbers):

This study was approved by Ethics Review Committee of CMH Lahore Medical College and Institute of Dentistry.

## Supplemental Information

Supplemental information for this article can be found online at http://dx.doi.org/10.7717/peerj.698#supplemental-information.

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
