# Peer review of "Public stigma associated with mental illnesses in Pakistani university students: a cross sectional survey"

_PeerJ, doi:10.7717/peerj.698_

## Round 0.1 · original submission · Minor Revisions

Dear Authors, Please take heed of the comments of the 3 peer reviewers who have given excellent feedback on how to improve the manuscript which includes statistics (but all the p values are represented as significant by placing stars. Please do check the representation of significance in other tables too). We hope to receive a revised manuscript with the corrections completed as soon as possible.

·

Basic reporting

Table1: wrongly represented. None of the p values are significant (<0.05), but all the p values are represented as significant by placing stars. Please do check the representation of significance in other tables too.

Experimental design

"No Comments".

Validity of the findings

Line 74: you have mentioned modified version of the CAMI, but how the scale is modified has not been mentioned throughout. Better to use adapted and translated (if any).
Line 115-116: percentage doesn’t add up to hundred.
I am bothered about the analysis. Results shown in tables need to be confirmed for their significance as I mentioned in my comment under basic reporting

Additional comments

Well written except for the result part. Can be considered for publication after making the results clear. Please do show the p values in all the tables.
Lines 59-62: objective number one is assessing two prevalence’s. Ideally one objective should capture only one idea. Also objective three shows repetition of objective one.

·

Basic reporting

The author used the word "stigmas" many times without explaining what are types of the stigmas he mean. So, I believe he should mention types of stigma like "public and self stigma" as an introduction to the reader.

Experimental design

No comments

Validity of the findings

No comments

·

Basic reporting

- Table 1, 3rd column P value: yet the asterix indicate another P value which is confusing
- table 1, 2nd column: what does number (1) mean?
- table 2, 2nd column: what does number (2) mean?
- referencing not according to format
- reference no.1: SD actually Scott DR
- another example is no.7: the authors should be written as Link BG, Struening EL, Rahav M, Phelan JC, Nuttbrock L. (authors should check with pubmed or similar database)
- it is good, if author can mention the 2 type of stigma ie self- and public stigma and CAMI measures public stigma

Experimental design

research questions clearly stated, relevant and meaningful

Validity of the findings

findings are valid.

Additional comments

- "stigmas" is should be replaced with stigma. If the author want to use the plural form, it should be stigmata
- line 213: delete "to"

---

## Round 0.2 · accepted · Accept

Dear Authors,Congratulations on the revised manuscript being accepted.Please edit these small mistakes as well:Only very minor inconsistency in the references
- reference number 4 has &
- references 20, 21, 22 have "and" before last author
Thanking You

·

Basic reporting

The article has met my previous comment. Also, the authors mentioned a hint about the public stigma and its effects on seeking help from mental health professionals.

Experimental design

No comments.

Validity of the findings

No comments.

·

Basic reporting

Only very minor inconsistency in the references
- reference number 4 has &
- references 20, 21, 22 have "and" before last author

Otherwise, I am satisfied with the revised manuscript.

Experimental design

Correction done and I am satisfied

Validity of the findings

I am satisfied with the revised manuscript